# Reproductive strategy of Delta Smelt *Hypomesus transpacificus* and impacts of drought on reproductive performance

**Tomofumi Kurobe**[1]*, **Bruce G. Hammock**[1], **Lauren J. Damon**[2], **Tien-Chieh Hung**[3], **Shawn Acuña**[4], **Andrew A. Schultz**[5], **Swee J. Teh**[1]

1 Department of Anatomy, Physiology, and Cell Biology, School of Veterinary Medicine, University of California Davis, Davis, California, United States of America, 2 California Department of Fish and Wildlife, Stockton, California, United States of America, 3 Department of Biological and Agricultural Engineering, University of California, Davis, Davis, California, United States of America, 4 Metropolitan Water District of Southern California, Sacramento, California, United States of America, 5 Science Division, U.S. Bureau of Reclamation Bay-Delta Office, Sacramento, CA, United States of America

* tkurobe@ucdavis.edu

**Data Availability Statement:** All relevant data are within the paper and its Supporting Information files.

## Abstract

Understanding reproductive biology and performance of fish is essential to formulate effective conservation and management programs. Here, we studied reproductive strategies of female Delta Smelt *Hypomesus transpacificus*, an endangered fish species in the State of California, the United States, focusing on (1) better understanding their distribution pattern during the winter and spring spawning season at very fine scale to predict their possible spawning grounds and (2) assessing impacts of a recent, severe drought on their reproductive performance. We formulated our hypotheses as follows; (1) female Delta Smelt migrate to particular locations for spawning so that mature females can be frequently found in those locations throughout the spawning season and (2) reproductive performance of individual female fish declined during the drought. To test the first hypotheses, we analyzed relationships between water quality parameters and maturity/distribution pattern of Delta Smelt. Salinity better explained the distribution pattern of Delta Smelt at subadult and adult stages compared with water temperature or turbidity. Although there are some freshwater locations where mature Delta Smelt can frequently be found during the spawning season, Delta Smelt at the final maturation stage (Stage 5: hydration) and post spawners appeared to be widespread in the area where salinity was below 1.0 during the spawning season. Therefore, Delta Smelt could theoretically spawn in any freshwater locations, with more specific spawning requirements in the wild (e.g., substrate type and depth) still unknown. Delta Smelt, which experienced dry and critically dry conditions (the 2013 and 2014 year-classes), showed smaller oocytes, and lower clutch size and gonadosomatic index compared with the fish caught in a wet year (2011 year-class) at the late vitellogenic stage (Stage 4 Late), suggesting reproductive performance was negatively affected by environmental conditions during the drought.

**Funding:** This work was supported by a grant to SJT and TK (U.S.Bureau of Reclamation R17AC00129). Partial support was provided by grants to SJT from the California Department of Fish and Wildlife Ecosystem Restoration Program E1183004 and U.S. Geological Survey G12AC20079 and G15AS00018 (Erwin Van Nieuwenhuyse program manager). The funders had no role in study design, data collection and analysis, decision to publish, or preparation of the manuscript.

**Competing interests:** The authors have declared that no competing interests exist.

## Introduction

Fish species are facing extinction all over the world. Anthropogenic activities are the primary cause of fish population declines, including overfishing and habitat destruction [1, 2]. In populations declining due to habitat destruction, habitat protection and restoration are tools for successful fish population recovery. For example, the removal of dams improved fish passages, created wetlands as more natural flow regimes were restored (e.g., seasonal flooding), and subsequently increased diversity of fish species [3–5]. However, in most cases, habitat restoration and protection are challenging as they often require reconciliation with anthropogenic activities.

In Northern California, USA, the fresh water entering the Sacramento and San Joaquin River Delta (hereafter the Delta), upstream from the San Francisco Estuary (SFE), is mainly supplied by rainstorms in winter, snowmelt from the Sierra Nevada mountains in early summer, and water released from reservoirs in late summer and fall, providing habitats for resident fishes (Fig 1A). Fresh water from the Delta is also exported to the Central Valley, Southern California, and other surrounding locations to irrigate millions of hectares and for use by municipalities. The freshwater exports from the Delta affect the availability of freshwater habitats and influence the location of the low salinity mixing zone in the SFE in which saltwater and freshwater create brackish-water habitat [6, 7].

The population sizes of pelagic fish species in the SFE-Delta have been declining since at least 1980 [9, 10]. The causes of the fish population declines are still unknown because there are numerous ecological issues in the SFE-Delta, such as habitat destruction by levee construction, changes in physical water parameters due to the water exports, invasive organisms (e.g., clams), low phytoplankton and zooplankton abundances, blooms of harmful algae, and contaminants released from urbanized areas and agricultural lands [11–15]. It is plausible to think that a combination of these ecological issues is contributing to the decline of fish populations [16]. Adding to these long-term changes was a severe drought from 2012–2015, which peaked in severity during 2014 and 2015. Precipitation was far below average and air temperature was historically high, resulting in reduced freshwater inflow, saltwater intrusion, and elevated water temperatures in Northern California [17, 18].

Reduced freshwater inflow to the SFE may be one of the main factors that negatively affects fitness and performance of endemic fish species. Bennett [11] discussed possible relationships between abundance of Delta Smelt and salinity in the Suisun Bay/Marsh (Fig 1B). Later, Jassby et al. [19] and Feyrer et al. [7] reported that the amount of freshwater inputs and location of the low salinity mixing zone influences the biotic resources for fishes, particularly the abundance and distribution of prey items such as copepods and mysids. In addition, Hammock et al. [20] reported that freshwater exports and the invasion of the clam *Potamocorbula amurensis* largely explain the current low concentrations of chlorophyll *a* in the SFE.

Delta Smelt (*Hypomesus transpacificus*) is endemic to the SFE-Delta and is federally listed as a threatened species [21]. It is an annual, multi-spawning fish species belonging to the family Osmeridae. Juvenile and sub-adult stages of fish can occur in the summer and fall, respectively, and spawning predominantly occurs in the spring [11, 22]. Delta Smelt has three major life-history phenotypes: freshwater resident, brackish-water resident, and semi-anadromous [23, 24]. The relative abundance of each life history phenotype varied inter-annually with the migratory phenotype being most common in every year, but not always dominant, and the brackish-water resident the least common. In 2011, the majority of Delta Smelt (81%) were semi-anadromous, with fish rearing in the low salinity mixing zone (salinity 1 to 6), and migrating back to freshwater regions during the spawning season [24]. Delta Smelt are thought to initiate an annual spawning migration that appears to begin immediately following the 'first

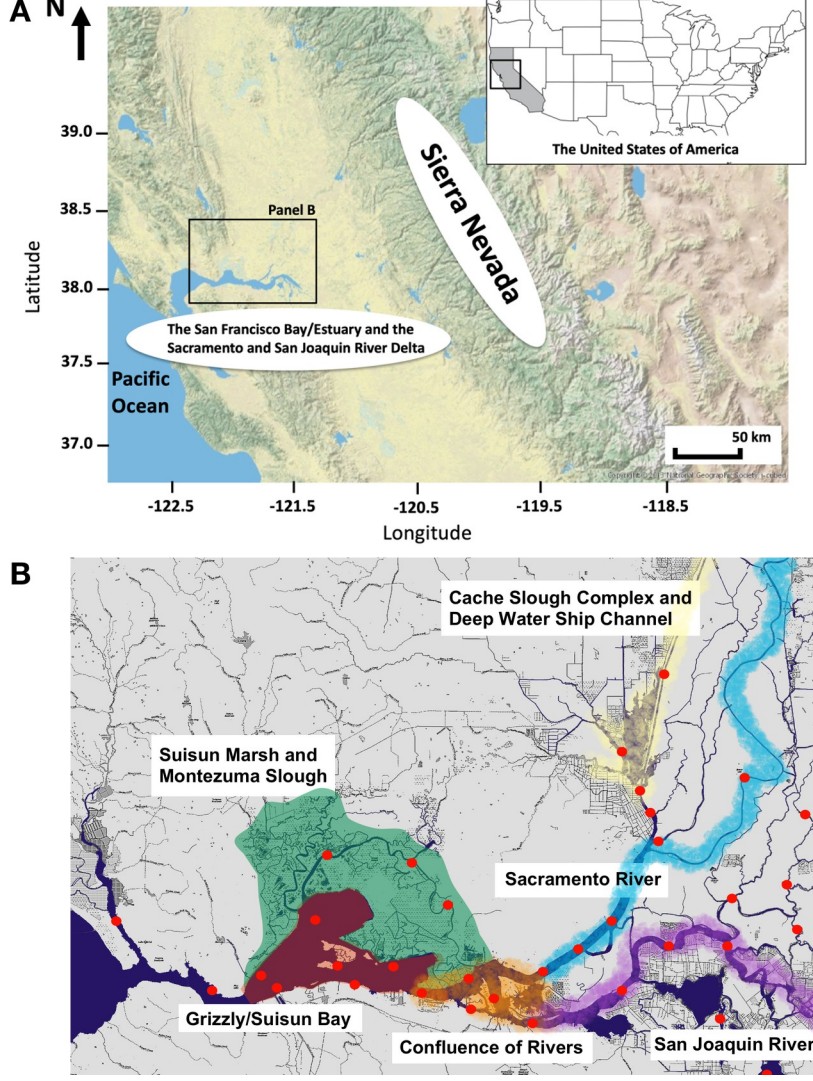

**Fig 1.** Map of Northern California, USA (Panel A) and the sampling stations (red points) for a long-term monitoring survey, Spring Kodiak Trawl Survey, conducted by the California Department of Fish and Wildlife (Panel B). The base maps were downloaded from the U.S. Geological Survey (https://apps.nationalmap.gov/viewer/). The sampling stations in the Panel B were plotted using the R package 'ggmap' [8].

flush', the arrival of turbid water from land runoff mobilized by the first major winter rainstorm [25, 26].

There are several papers regarding migration and distribution of Delta Smelt during the spawning season in the SFE-Delta. Sommer et al. [25] reported spawning migration of Delta Smelt to upstream regions such as the Cache Slough Complex (Fig 1B) with an emphasis on migration rates and timing of maturation. Similarly, seasonal distribution patterns of Delta Smelt by life stage was thoroughly studied by Murphy and Hamilton [27] and the authors reported that relative densities of Delta Smelt in Suisun Bay and the confluence region diminished while high densities were found in Montezuma and Cache Slough Complex during the spawning season. Bennett and Burau [26] concluded that Delta Smelt exploit tidal action to migrate to upstream spawning habitat in the northern Delta (e.g., Cache Slough Complex). Furthermore, Hammock et al. [28] studied the causes and consequences of migration from the

viewpoint of foraging and food availability in the SFE-Delta, reporting increased stomach full-ness in brackish-water regions during fall, winter, and spring compared to freshwater regions. However there still are unknowns, such as environmental factor(s) that affect regional to finer scale distribution of Delta Smelt during the spawning season. Turbidity is one possible migra-tion cue as Delta Smelt distributions are associated with turbid water [12, 25, 26, 29]. In addi-tion, it is unclear whether there are particular spawning grounds, or whether Delta Smelt spawn at any locations where water quality parameters are suitable for mature fish.

Another unknown is whether a severe, recent drought in California from 2013–2015 impacted the reproductive performance of Delta Smelt. Feyrer et al. [7] forecasted that reduced freshwater flow, particularly due to drought, can decrease the abundance of Delta Smelt via changes in the location of the low salinity mixing zone where the majority of semi-anadro-mous Delta Smelt occur as juveniles/sub-adults. More specifically, if (1) migratory form of immature Delta Smelt (e.g., juvenile and subadult stages) are predominantly distributed in the low salinity mixing zone and (2) the low salinity mixing zone is located in suboptimal regions for the growth of Delta Smelt (i.e., poor food availability) due to the changes in water flow, mature fish may exhibit a reduced reproductive performance due to the decreased energy reserves. Droughts associated with climate change are predicted to increase in frequency, dura-tion, and intensity [30, 31]. Given the relationships among freshwater flow, fish distribution, and habitat availability and accessibility, it is crucial to assess the impact of the recent drought on Delta Smelt reproduction at the highest resolution.

In addition to habitat loss, elevated water temperature associated with the drought is another concern for the reproductive success of Delta Smelt [32–34]. Fishes are ectotherms and many species, especially temperate ones, evolved to cope with seasonal changes in water temperatures for their reproduction [35]. Water temperature can affect various reproductive processes of fishes, such as gamete development and maturation [36], therefore it was very likely that the elevated water temperature during the drought in California affected Delta Smelt reproduction. Damon et al. [32] found that while clutch fecundity at length was consis-tent among years, the spawning window diminished during drought years due to elevated tem-peratures in the SFE, resulting in Delta Smelt having lower annual fecundity (i.e., total number of eggs released per female per year) since each female produced fewer clutches. Temperature may have also affected fish size, resulting in reduced clutch size [32].

The main objectives of this study are to improve understanding of the reproductive strategy of female Delta Smelt and to assess the impact of drought on their reproduction by investigat-ing fish distribution and a suite of reproductive endpoints. We focused on females because their reproductive performance is more directly associated with recruitment success than males in other fishes [37]. We formulated our hypotheses as follows: (1) female Delta Smelt migrate to particular locations for spawning so that mature females can be frequently found in particular locations throughout the spawning season and (2) reproductive performance of individual female fish declined during the drought.

To examine the hypotheses, we analyzed relationships between water quality parameters and maturity/distribution of female Delta Smelt. Females were collected across the SFE-Delta via long term monitoring surveys conducted by the California Department of Fish and Wild-life (CDFW), collected from winter 2011 through spring 2015. To obtain high resolution sex-ual maturity data, ovaries of Delta Smelt were staged by histological examination following Kurobe et al. [38]. We first investigated relationships between water quality parameters (i.e., salinity, water temperature, and turbidity) and maturity of female Delta Smelt. We further examined the distribution of female Delta Smelt at each stage during the spawning season for each year. To assess the impact of drought on reproductive performance of female Delta Smelt, we compared reproductive metrics of mature fish at the late vitellogenic stage: oocyte

areas, clutch size, and gonadosomatic index, across four years representing a wide range in water year types. Together, the three reproductive metrics represent reproductive fitness of each individual, including egg quality (oocyte area), number of eggs that individual female fish produce per spawning event (clutch size), and the mass of ovary relative to body weight (gonadosomatic index). Finally, the timing of maturation between wet and dry/drought years was also compared among different year-classes.

## Materials and methods

### Ethics statement

A California Endangered Species Act (CESA) Memorandum of Understanding (MOU) was made and entered into by and between Dr. Swee Teh of the University of California, Davis (permittee) and the CDFW. The purpose of this CESA MOU was to authorize the permittee to obtain and possess Delta Smelt (*Hypomesus transpacificus*) collected by the CDFW Interagency Ecological Program for scientific purposes pursuant to Fish and Game Code (FGC) 2081 (a). This study was approved by the University of California, Davis Institutional Animal Care and Use Committee and followed the experimental protocol for Animal Care and Use protocol #19872.

### Fish sampling

Delta Smelt adults were collected during a long-term survey: Spring Kodiak Trawl (January through April or May) conducted by the CDFW [39–42]. Sampling stations for the Spring Kodiak Trawl are shown in Fig 1B. All the details for the sampling stations (e.g., GPS coordinates) are available at the CDFW website (https://www.dfg.ca.gov/delta/projects.asp?ProjectID=SKT). Female Delta Smelt collected from 2011 (n = 461), 2012 (n = 123), 2013 (n = 174), and 2014 year-classes (n = 74) were used for the analyses. Year-classes were defined by hatch year. For example, fully matured fish in spring 2012 were considered the 2011 year-class. Delta Smelt are generally annual fish and second year fish are rarely observed [32]. A summary of Spring Kodiak Trawl for 2011 through 2014 year-classes, showing number of stations sampled at each region for each month, is in S1 Table. Field sampling logistics (i.e., number of stations that field sampling was performed) slightly differed for each month.

In the field, Delta Smelt were sacrificed by pithing and dissected to provide real-time information on their maturity status. Care was taken to minimize egg loss. After dissection, Delta Smelt were individually wrapped in aluminum foil with an identification number, and then flash-frozen in liquid nitrogen on sampling boats. Liquid nitrogen dewars containing fish samples were brought back to the Aquatic Health Program, University of California, Davis where Delta Smelt samples were stored in liquid nitrogen until processing [38, 43].

### Fish dissection and calculation of somatic condition factor and gonadosomatic index

Individual Delta Smelt were removed from liquid nitrogen and fork length and body weight were measured while the fish were still frozen (S2 Table). The ovaries were excised once the fish were partly defrosted. The ovaries were weighed and partitioned into two portions by a scalpel; one portion (~60%) was fixed in 10% phosphate-buffered formalin for histology and the remainder (~40%) was stored at −80˚C for clutch size estimation. Somatic condition factor ($K_s$) and gonadosomatic index ($GSI$) were calculated by the formulas $K_s = (W_t - W_g) / L_f^3 \times 100$, where: $W_t$ = total weight (g); $W_g$ = gonadal weight (g); $L_f$: fork length (cm) and $GSI = (W_g / W_t) \times 100$ [38, 44].

## Water quality parameters and hydrologic classification

Physical water quality parameters including water temperature, turbidity, conductivity, and Secchi depth were measured from the boats [41]. The identification number on the aluminum foil was used to associate individuals with catch location, date, and water quality.

Hydrologic classifications of water years were obtained from the California Data Exchange Center, California Department of Water Resources (http://cdec.water.ca.gov/reportapp/javareports?name=WSIHIST). All the details for the hydrologic classification of water years can be found on the website. Based on the Sacramento Valley Index, 2011 was a wet year, followed by a below normal year (2012), a dry year (2013), and a critically dry year (2014).

## Maturity of female Delta Smelt

Histological examination was used to stage maturity of female Delta Smelt based on morphological changes in ovaries (S2 Table) [38]. Briefly, ovaries fixed in 10% phosphate-buffered formalin were embedded in paraffin blocks and sectioned to a thickness of 3 μm using a microtome. Each section was mounted on a glass microscope slide and stained with hematoxylin and eosin solution [38, 43]. Based on the histological characterization of ovaries (i.e., majority of the most advanced oocytes in ovaries), fish were categorized into six major stages: immature stage (Stage 1 and 2), cortical alveolus stage (Stage 3), vitellogenic stage (Stage 4), final maturation stage (hydration stage, Stage 5), and post-spawners (Stage 6). Maturity at Stage 3 and 4 were further divided into three sub-stages, Early, Middle, and Late, based on the abundance of cortical alveoli in Stage 3 oocytes and egg yolk bodies in Stage 4 oocytes. Criteria for each stage designation are further described in our previous paper [38].

## Area of oocytes measured by histology

The cross-sectional areas of oocytes ($mm^2$) were obtained using histological images of ovaries (S2 Table). The average oocyte size and standard deviation were obtained based on measurements from approximately 10 oocytes randomly selected from each ovary [38]. Delta Smelt is a multiple spawner and immature oocytes can be found in their ovaries even during the spawning season [32, 38]. Therefore, only oocytes at the most advanced stage within each ovary were used for measuring oocyte area. To ensure that oocytes sectioned roughly in half were measured for area, only the oocytes with a visible nucleus transection were chosen for measurement [45]. ImageJ software ver. 1.8.0_112 was used for the image analysis [46].

## Clutch size

A portion of the ovary was used for estimating clutch size. Oocytes were dispersed and counted in 1× phosphate-buffered saline using a dissecting microscope, and clutch size (*C*) was calculated using this equation: $C = O_{portion} \times (W_{intact} / W_{portion})$, where *C* is the clutch size, $O_{portion}$ is the count of oocytes in the portion of ovary, $W_{intact}$ is the weight of the intact ovary, and $W_{portion}$ is the weight of the portion of the ovary used for counting oocytes (S2 Table).

## Data analysis

**Relationships between maturity of female Delta Smelt and water quality parameters.** To investigate relationships between maturity of female Delta Smelt and water quality parameters (i.e., salinity, water temperature, and turbidity), boxplots were made using the water quality data at stations where female Delta Smelt were collected. Additional bar graphs were made to depict median absolute deviation of water quality parameters at each maturity stage.

**Distribution pattern of female Delta Smelt during spring spawning season.** To examine our first hypothesis (i.e., female Delta Smelt migrate to particular locations for spawning so that mature females can be frequently found in particular locations throughout the spawning season), we visualized spatiotemporal distribution patterns of female Delta Smelt in the 2011–2014 year-classes at each major maturity level. Fish catch data were visualized by bubble map charts using Microsoft® PowerPoint for Mac (ver. 16.16.17).

In addition, historical data from the CDFW Spring Kodiak Trawl Survey were obtained from the CDFW website to assess spatiotemporal distribution patterns of female Delta Smelt for periods when the fish species was more abundant (CDFW Spring Kodiak Trawl: http://www.dfg.ca.gov/delta/projects.asp?ProjectID=SKT). Two objectives of the CDFW Spring Kodiak Trawl Survey are to determine the distribution and relative abundance of adult Delta Smelt in the SFE-Delta and to monitor their gonadal maturation on a monthly basis to determine when and where spawning is likely to occur or is occurring [42]. In the CDFW Spring Kodiak Trawl Survey, maturity of female Delta Smelt was classified into six stages by gross examination of their ovaries: developing stages (Stage 1 and 2), near-ripe (Stage 3), ripe (Stage 4), atretic (Stage 5), and post spawn (Stage 6) [32].

**Comparison of female Delta Smelt maturity among the 2011–2014 year-classes during the spawning season.** Maturity of female fish were scored by histological examination as described above (Section "Maturity of female Delta Smelt"). Stacked bar graphs showing the relative abundances of each maturity level for each month for the four-year classes were made using the R package ggplot2 ver. 3.2.1 [47, 48].

**Comparison of reproductive performance of female Delta Smelt between wet and dry/drought years.** To test our second hypothesis that reproductive performance of female Delta Smelt was lower during the dry and critically dry years, we performed one-way ANOVAs. The ANOVAs compared area of oocytes, clutch size, gonadosomatic index, and somatic condition factor among 2011–2014 year-classes [48]. ANOVA was chosen since the datasets are normally distributed (Shapiro-Wilk Test $P > 0.05$). Homogeneity of variance was assessed by Levine's Test, and were $\log_{10}$-transformed as necessary to meet the assumption. A *post-hoc* Tukey Honestly Significant Difference Test was performed when statistically significant differences were detected by ANOVA. The analyses were performed only for fish at the late vitellogenic stage (Stage 4 Late) since the fish at this stage have mature oocytes and are about to spawn, and therefore better represent reproductive performance of spawning females compared with earlier stages [38]. We did not include fish at the final hydration stage (Stage 5) because fish at this stage are rare (comprise <1% of adult female Delta Smelt in this study), and may have been actively spawning at capture, affecting clutch size and GSI.

We also analyzed dynamic changes of water temperature that a majority of Delta Smelt in the 2011–2014 year-classes were likely experiencing in the wild: from April 1st, 2011 (larvae in the 2011 year-class) through March 31st, 2015 (adults in the 2014 year-class). The water temperature data collected at time of capture, along with other water quality parameters were downloaded from the CDFW FTP server (ftp://ftp.wildlife.ca.gov/). Water quality data from two surveys, 20-mm Survey (March-July) and Spring Kodiak Trawl Survey (January-April) were pooled [41] and a subset of the data was prepared based on two criteria: regions and salinity range. Only the water temperature data in the four regions, (1) Suisun Marsh/Bay and Montezuma Slough, (2) Confluence of the Sacramento and San Joaquin rivers, (3) Sacramento River, and (4) Cache Slough Complex, were used for the analysis. Data from other regions such as San Joaquin River were not included, nor was data from high salinity stations (salinity >6.0).

## Results

### Water quality parameters and maturity of female Delta Smelt

The pooled data from the 2011–2014 year-classes showed unique relationships between maturity of female Delta Smelt and water quality parameters. The median salinity values decreased as female Delta Smelt matured (Fig 2A). At the early cortical alveolus stages (Stage 3 Early and Middle, grouped), 78% of fish were found at a wide range of salinities (salinity < 6.0). In contrast, Delta Smelt at the late cortical alveolus stage (Stage 3 Late) and later stages were collected at lower salinities as their maturity level advanced. Approximately 77% of fully matured fish (Stage 4 Late and 5) and post-spawners (Stage 6) were found in fresh water (salinity <0.5). The median values of water temperature decreased from the immature stage (Stage 2) to the late cortical alveolus stage (Stage 3 Late) and elevated for the vitellogenic stage (Stage 4) to post spawners (Stage 6; Fig 2B). Fish maturity increased with turbidity except for the immature stage (Stage 2), which occurred at elevated turbidity (Fig 2C). Fish at the final hydration stage (Stage 5) showed the highest median turbidity of 33 Nephelometric Turbidity Units (NTU); $25^{th}$ and $75^{th}$ percentiles were 23 and 39 NTU, respectively. Note that the sample size at the immature (Stage 2) and final hydration stages (Stage 5) were relatively low compared with fish at other stages in our data. During the four-year study, we found only 13 and 14 fish at Stage 2 and 5, respectively (Fig 2).

For the ease of data presentation, fish with extreme turbidity values (>150 NTU) are not included in Panel C. The original figure including those fish with extreme turbidity values can be found in S1 Fig.

The comparison of variabilities in salinity, water temperature, and turbidity across reproductive stages revealed a contrast; for salinity, the variability gradually decreased as Delta Smelt maturity level advanced, while such clear trend was not observed for water temperature

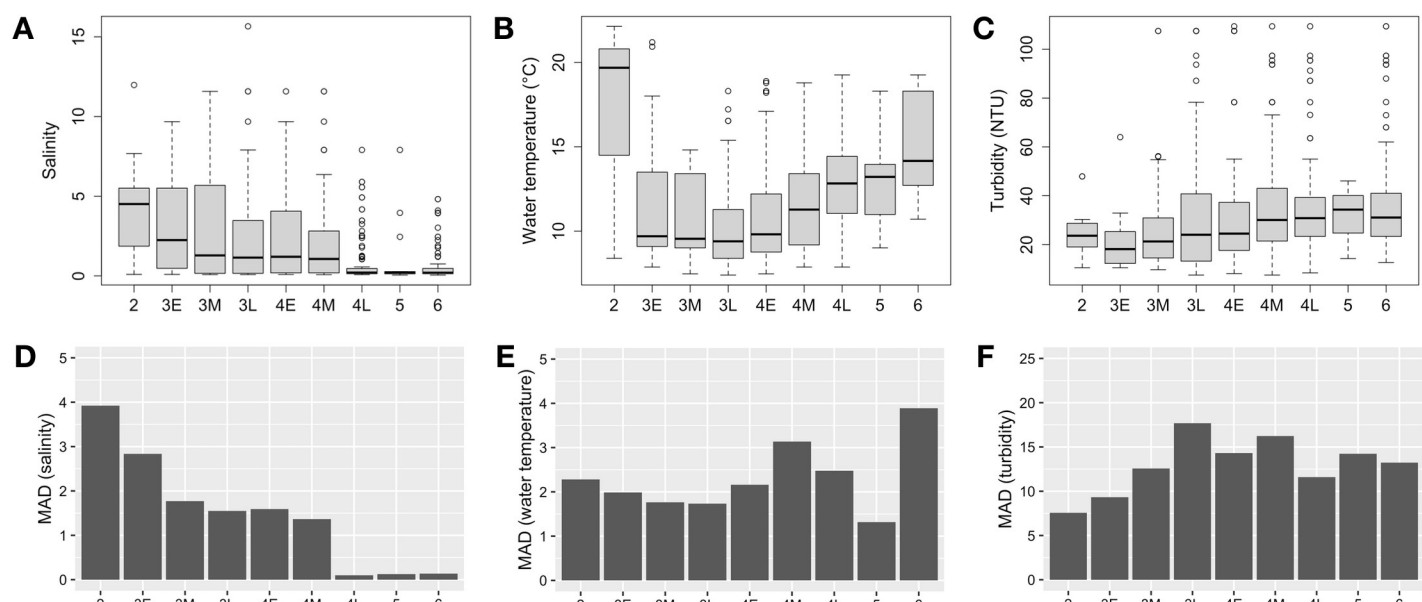

**Fig 2.** Summary of salinity (A), water temperature (B), and turbidity values (C) at each reproductive stage of female Delta Smelt, 2011–2014 year-classes, and corresponding median absolute deviations (MAD) (D-F). Maturity of female fish was scored based on the gonadal histological features (Stage 1&2: immature stages, Stage 3: cortical alveolus stage, Stage 4: vitellogenic stage, Stage 5: final maturation stage or hydration stage, and Stage 6: post spawners [38]). Number of fish at each reproductive stage are as follows: Stage 2 (n = 13), Stage 3E (n = 36), Stage 3M (n = 64), Stage 3L (n = 200), Stage 4E (n = 112), Stage 4M (n = 121), Stage 4L (n = 156), Stage 5 (n = 14), and Stage 6 (n = 117).

or turbidity (Fig 2D–2F). We therefore selected salinity for further investigation at a finer scale because salinity showed the clearest relationship with maturation of Delta Smelt.

## Geographical distribution of female Delta Smelt during spawning season

Data from the 2011–2014 year-classes suggested that (1) a majority of mature female Delta Smelt were found at salinities below 1.0 after spawning migration and (2) there were some locations where mature fish were frequently found during the spawning season (i.e., late spring), such as Cache Slough Complex and Suisun Marsh/Montezuma Slough, (3) however there was plasticity in the distribution of Delta Smelt, as fish appeared to be widespread in the area where salinity was below 1.0 during the spawning season. In January 2014, sub-adult female Delta Smelt in the 2013 year-class exhibited a wide geographical distribution ranging from a freshwater area (the Cache Slough Complex) to a brackish-water area (Suisun Marsh/ Montezuma Slough) where salinity was approximately 6.0 (Fig 3I). This pattern changed in March 2014 when most females were at the vitellogenic stage (Stage 4); all the fish were found in the locations where the salinity was below 1.0, including the Cache Slough Complex, Suisun Marsh/Montezuma Slough, and Sacramento River (Fig 3K). By the following month, fish were collected mostly in the Cache Slough Complex, with only a few fish found in other freshwater areas (Fig 3L). A similar progression of distribution was observed in the historical data, particularly in 2001 through 2003 year-classes (S2 Fig). The 2001 through 2003 as well as 2011 year-classes showed a wider distribution in April and/or May throughout the SFE-Delta, but mature females were still mostly found at salinities below 1.0 (Figs 3D and S2). The data for the 2014 year-class can be found in S3 Fig.

Similar to fish at the vitellogenic stage (Stage 4), a majority of post spawners (Stage 6) were found in freshwater areas at salinities below 1.0, however a wide distribution pattern was observed in April 2012 (Fig 3D). Post spawners were found across the SFE-Delta, including Grizzly/Suisun Bay and the confluence of the Sacramento and San Joaquin rivers, as the freshwater area expanded due to the intense winter rainstorms. Similarly, most of fish at Stage 5, characterized as fully mature fish with hydrated eggs, were also observed in freshwater areas in March and April 2012 (the Sacramento River and Cache Slough Complex; Fig 3C and 3D). However, two fish at the stage were collected in brackish water areas in March 2013 (Fig 3G) where salinity was 1.0–6.0 (Grizzly/Suisun Bay and Suisun Marsh/Montezuma Slough).

## Comparison of female Delta Smelt maturity among the 2011–2014 year-classes during the spawning season

A prominent difference is apparent between 2011 and other year-classes; female Delta Smelt in the 2013 and 2014 year-classes matured earlier than in the 2011 year-class (Fig 4). In the 2013 and 2014 year-classes, Delta Smelt at the vitellogenic stages (Stage 4 Early, Middle, and Late) were dominant in February, accounting for over 90% of the females (Fig 4C and 4D). In contrast, only 62% of the fish collected in February 2011 were at the vitellogenic stage (Fig 4A).

We found over 15% of fish at the late cortical alveolus stage (Stage 3 Late) in March and April in the 2012 year-class (Fig 4B). Those fish were very likely to be post-spawners although there were no signs of spawning such as presence of post-ovulatory follicles in the histological sections. The cortical alveolus stage (Stage 3) is characterized by the presence of cortical alveoli, which can be found in the subadult stage of Delta Smelt [38]. The mean fork length of the fish at late cortical alveolus stage (Stage 3 Late) collected in March and April (77.0 mm) was significantly longer than the same stage of fish collected in January (67.7 mm) ($t_{[24]} = 4.79$, $P < 0.001$) and was similar to the one at Stage 6 fish (post spawners, mean fork length: 76.1 mm) collected in March and April in the 2012 year-class ($t_{[23]} = 0.42$, $P = 0.68$).

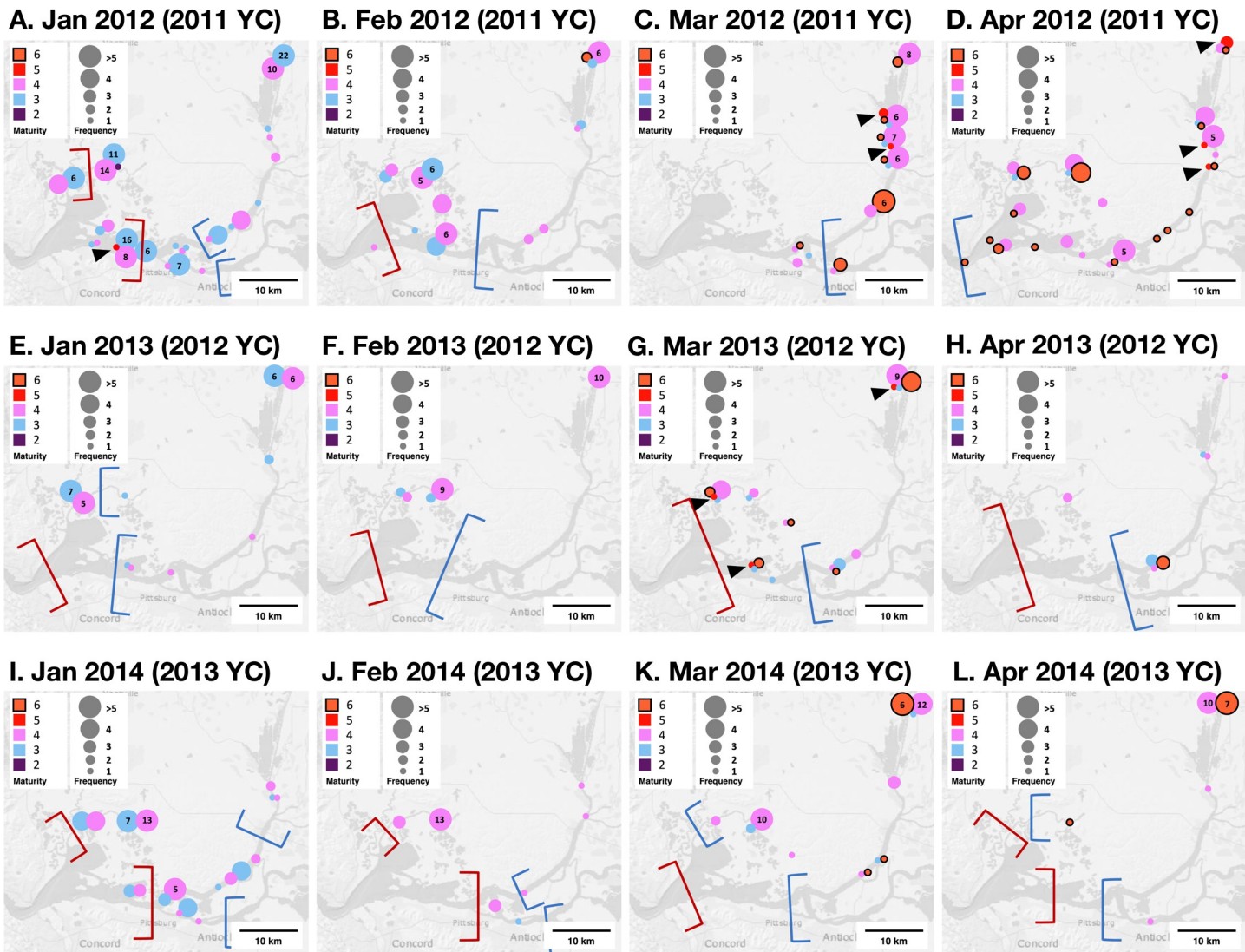

**Fig 3.** Maturity and distribution pattern of female Delta Smelt, 2011 (January-April 2012, Panels A-D), 2012 (January-April 2013, Panels E-H), and 2013 year-classes (YC) (January-April 2014, Panels I-L). Fish at the final hydration stage (Stage 5) are indicated by arrow heads. The blue and red brackets indicate a simplified salinity boundary for 1.0 and 6.0, respectively. The data from May were not included due to the very small sample numbers. The field sampling was not performed in the Grizzly/ Suisun Bay in March 2012 (Panel C, S1 Table). The data for the 2014 year-class can be found in S3 Fig. The base map was downloaded from the U.S. Geological Survey (https://apps.nationalmap.gov/viewer/).

## Comparison of oocyte area, clutch size, gonadosomatic index, and somatic condition factor among 2011–2014 year-classes

The comparison of oocyte areas revealed that there was no statistically significant difference in area of oocytes among the cohorts at the late vitellogenic stage (Stage 4 Late; Fig 5B; ANOVA, $F_{[3, 144]} = 1.9$, $P > 0.05$). However, clutch size was lower in 2013 and 2014 year-classes than 2011 year-class while the difference between 2011 and 2014 year-classes was not significant (Fig 5C, ANOVA, $F_{[3, 144]} = 7.3$, $P < 0.0005$). A similar trend was observed in gonadosomatic index; the 2013 and 2014 year-classes had lower values than the 2011 and 2012 year-classes (Fig 5D, ANOVA, $F_{[3, 144]} = 7.2$, $P < 0.001$). Significant differences were not observed among the 2011 through 2014 year-classes in somatic condition factor (Fig 5A, ANOVA, $F_{[3, 144]} = 1.5$, $P > 0.05$).

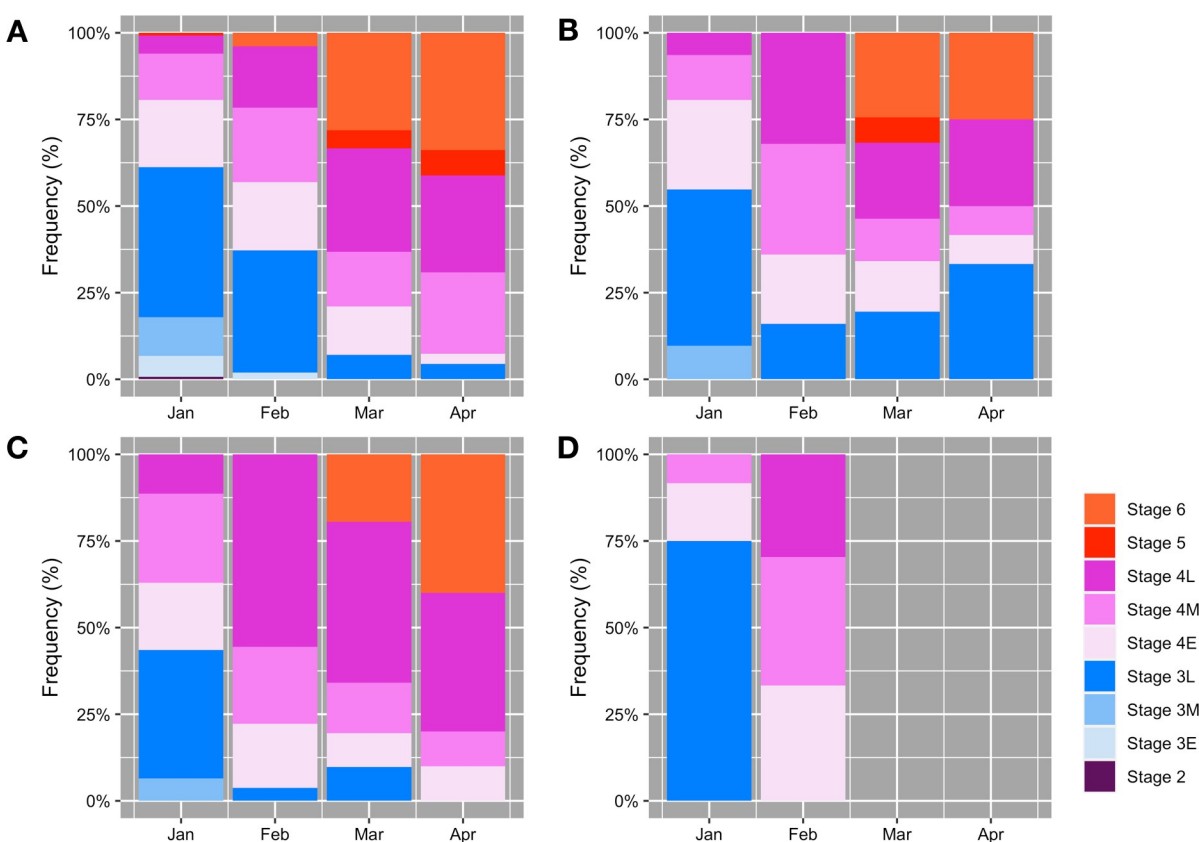

**Fig 4. Monthly changes of female Delta Smelt maturity levels for 2011–2014 year-classes.** Months with extremely low catches (n < 5) are not depicted in the figure (i.e., March and April in the 2014 year-class; Panel D).

## Discussion

Understanding the reproductive biology of imperiled fishes is one of the key elements in effective conservation and restoration of fish populations. Several previous studies reported migration, seasonal distribution patterns, and timing of spawning of Delta Smelt [6, 11, 24–26]. However, field-based data for environmental factors that affect distribution of Delta Smelt during the spawning season at regional to finer scales are still limited. In addition, the exact spawning ground of Delta Smelt is still unknown; it is unclear whether Delta Smelt spawn at any locations where water quality parameters are suitable, or whether there are particular regions for their spawning. Furthermore, little is known regarding how drought condition affects the reproductive performance of Delta Smelt. The data from this study provide insight into (1) regional and seasonal differences in maturity of Delta Smelt that can be further used to predict potential spawning ground and (2) impact of drought on reproductive performance.

Migration of Delta Smelt to brackish-water regions at early life stages appears to improve foraging success. Hammock et al. [28] reported that Delta Smelt collected from the brackish-water area in the fall through spring showed higher stomach fullness than those from the freshwater areas during the same period, while the opposite was true in the summer. Data from other smelts also supports the hypothesis that smelts migrate to brackish-water or saltwater environments for foraging and growth. For example, migratory populations of European Smelt (*Osmerus eperlanus*) were larger than those from non-migratory freshwater residential populations [49]. Similarly, anadromous or migratory Wakasagi (*H. nipponensis*) were larger than freshwater populations

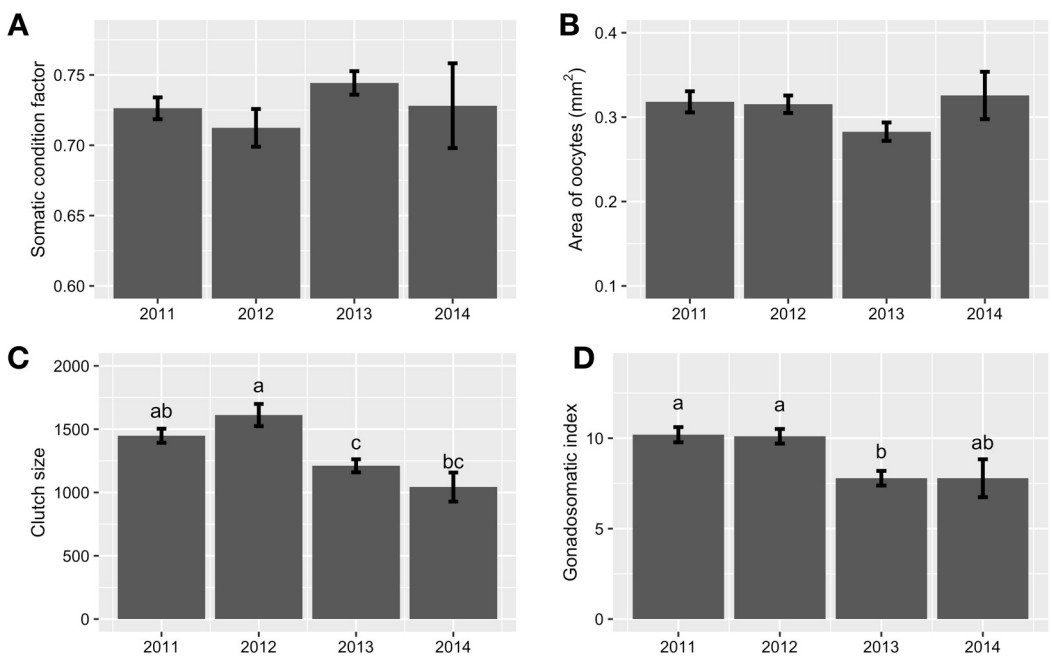

**Fig 5.** Comparison of somatic condition factor (A) and reproductive endpoints: area of oocytes (B), clutch size (C), and gonadosomatic index (D), among 2011–2014 year-classes. Delta Smelt collected in a wet (2011 year-class), below normal (2012 year-class), dry (2013 year-class), and critically dry year (2014 year-class), were used for the comparison. Only the fish at the late vitellogenic stage (Stage 4 Late) were used for the comparison. Different letters indicate statistically significant differences between year-classes. Error bars are ±SE.

[50]. These findings can explain the migration of Delta Smelt to brackish regions, however, it is still unclear why smelts migrate back to freshwater regions to spawn. Given that mature Delta Smelt (Stage 4 Late and 5) were predominantly found in freshwater regions, it is plausible to think that spawning in turbid freshwater environments is somehow beneficial. Hammock et al. [28] discussed the possibility that Delta Smelt spawn in freshwater regions to provide hatchlings and young fish better access to high summertime densities of mesozooplankton species. In other anadromous fishes, spawning in freshwater environments is thought to reduce predation risks [51]. For example, Arctic Charr (*Salvelinus alpinus*) and Brown Trout (*Salmo trutta*) showed five to eight times higher daily mortality rates at sea than in freshwater environments [52]. Another possibility is that spawning in freshwater regions reduces salinity stress, which can affect fertilization, yolk sac sorption, early embryogenesis, and larval growth [53, 54]. For example, elevated salinity decreased growth rate of Chinook Salmon (*Oncorhynchus tshawytscha*) and Rainbow Trout (*Oncorhynchus mykiss*) at larval and juvenile stages [55]. In addition, data form Romney et al. [56] suggest that environmentally relevant salinities (2–8) can possibly reduce fertilization success of Delta Smelt eggs. Turbid water in the freshwater environment associated with winter rainstorms may also be important for survival of early life stages of Delta Smelt. Hasenbein et al. [57] reported that larval Delta Smelt showed better survival and higher feeding rates in moderate turbidities (25–80 NTU). To culture Delta Smelt, turbidity is increased in larval stage by addition of phytoplankton because Delta Smelt at these life stages fail to feed in clear water [58–60]. Turbidity may help reduce predation risk as well. Delta Smelt cultured in turbid water (2.7 NTU) showed lower predation by Largemouth Bass (*Micropterus salmoides*) compared with ones in clear water (0.1 NTU) [61]. Similarly, Humpback Chub (*Gila cypha*) in turbid water significantly reduced predation by Rainbow Trout and Brown Trout (*Salmo trutta*) when cultured in turbid water as low as 25 Formazin Nephelometric Units [62].

The environmental driver(s) that trigger the onset of migration of Delta Smelt are still unknown, however it is very likely that changes in physical and/or physicochemical parameters due to winter rainstorm events are associated [25, 26]. Given the data from our study showing that (1) median salinity values decreased as fish matured, (2) mature female Delta Smelt and post spawners (Stage 4 Late, 5 and 6) were predominantly found in areas where salinity was less than 0.5 with very low variability, and (3) such clear trends were not observed in the relationships with water temperature or turbidity, salinity could better explain the distribution of Delta Smelt (Fig 2). This suggests that Delta Smelt may use freshwater influxes or its correlates, possibly associated with winter rainstorms, as a cue for their spawning migration. Fresh water is not the cue for the migration of the closely related species, Wakasagi, because spawning migration was observed even in an entirely freshwater environment, from a lake to adjacent inflowing rivers [63]. In Wakasagi, eggs were also found in sites at approximately 10 m depth, nearby underwater springs [64]. Nevertheless, turbidity seems to be still critically important for survival of early life stages of Delta Smelt, especially for foraging success as described above.

Otolith geochemistry has shown that semi-anadromous Delta Smelt migrate to freshwater regions to spawn (e.g., Cache Slough Complex) [24], however, spawning habitats and their distribution pattern after spawning migration, are still largely unknown. The data obtained in this study partly fill the knowledge gap. The distribution pattern observed in the 2013 and other year-classes such as 2001–2003 cohorts suggests that spawning habitat is more likely to occur in any freshwater locations, independent of geographic location (Figs 3 and S2). In April 2012 and March 2013, fully matured fish at Stage 5 were found along the Sacramento River and sites near the confluence of the Sacramento and San Joaquin River (Fig 3). Stage 5 is characterized as the final maturation stage and hydration takes place in oocytes [38]. Given that only a very small percentage of fish at Stage 5 were found in the four years of the study (14 out of a total of 832 fish), retention time of the oocytes at the final hydration stage (Stage 5) is likely to be very short. The short retention time of the hydration stage seems to be common in fishes. For example, in Northern Anchovy (*Engraulis mordax*) and Spotted Seatrout (*Cynoscion nebulosus*), the final maturation (hydration) and spawning can be completed within 24 hours [65, 66]. Although the retention time of Delta Smelt and other species in the family Osmeridae at the hydration stage (Stage 5) is still unknown, Delta Smelt at the stage can be an indicator of spawning areas assuming that the retention time is very short like other fish species. Given that the predicted upstream migration rates of Delta Smelt are 1.8–6.3 km per day [25], it is unlikely that the Delta Smelt at the final hydration stage (Stage 5) found in the Grizzly/Suisun Bay in April 2012 swam over 38 km of the distance and reached particular upstream freshwater locations such as Cache Slough Complex within the short retention time (Fig 3G).

The reproductive performance of fully matured Delta Smelt was relatively poor in the 2013 and 2014 year-classes (which experienced dry and critically dry years, respectively) compared with the 2011 year-class (which experienced a wet year) as indicated by lower clutch size and lower gonadosomatic index (Fig 5). Apparently, fish in the 2013 and 2014 cohorts produced less eggs than those in the 2011 cohort, but size of eggs were similar among the four cohorts (Fig 5). The cause of the poor reproductive performance in the 2013 and 2014 year-classes is still unclear, however it could be influenced by poor food availability during those years, stunting growth and reducing clutch fecundity [32]. The low salinity mixing zone (salinity between 1.0–6.0) was located at the confluence of the Sacramento and San Joaquin rivers in October 2013 and 2014 (Fig 6C and 6D). In contrast, the low salinity mixing zone included the Suisun Marsh and Montezuma Slough in October 2011, a region that is relatively rich in tidal wetlands (Fig 6A) [15]. Given that (1) the majority of Delta Smelt including both migratory and non-migratory populations (i.e., freshwater and brackish water residents) are distributed in

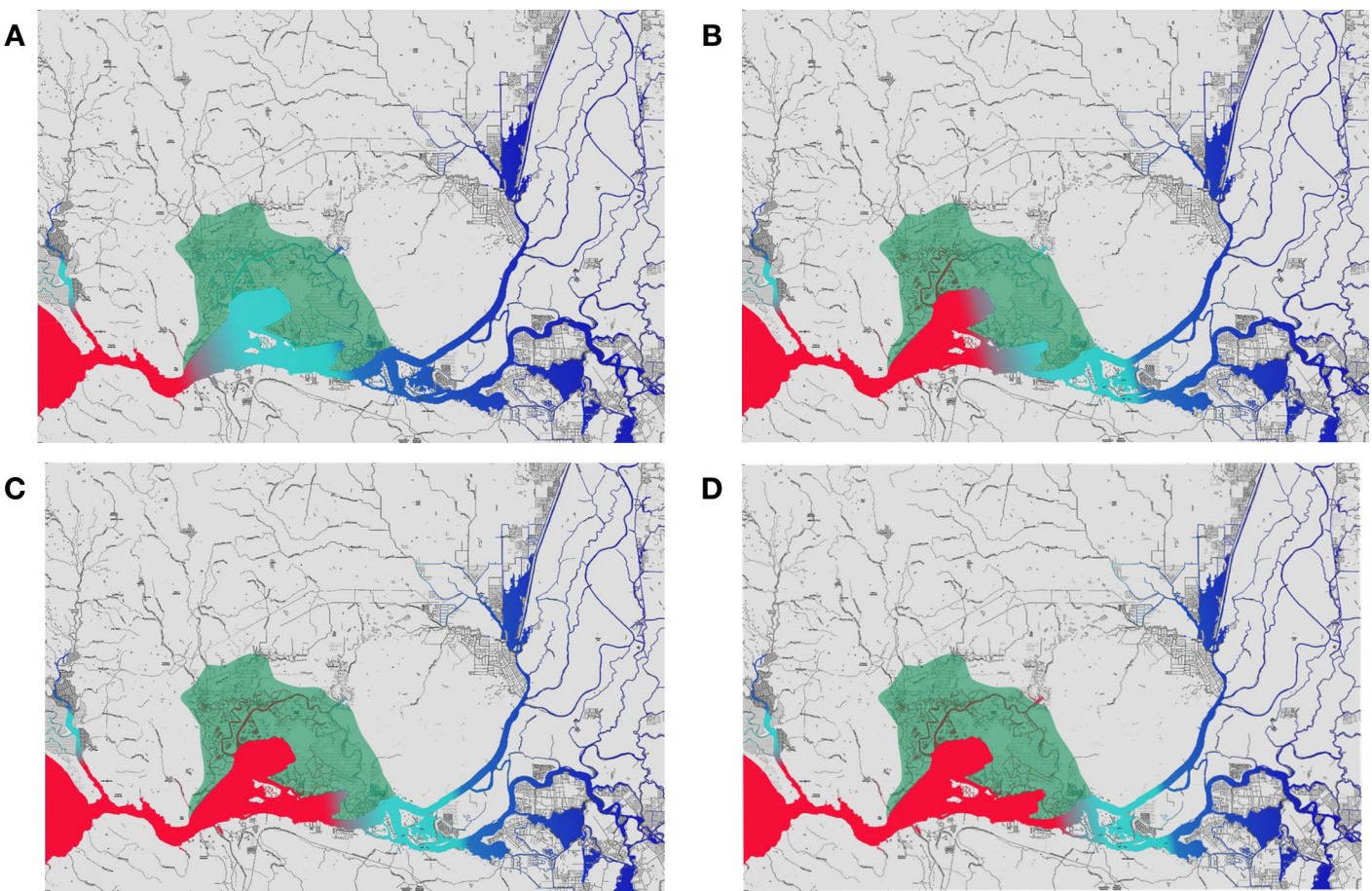

**Fig 6.** Comparison of area and location of the mixing zone (salinity 1.0–6.0) among 2011–2014 (A-D), October. The area with red, light blue, and dark blue color represent salinity >6.0, 1.0–6.0, and <1.0, respectively. The Suisun Marsh/Montezuma Slough area is indicated by green. Salinity data were obtained from the California Department of Fish and Wildlife FTP server (ftp://ftp.dfg.ca.gov/TownetFallMidwaterTrawl/). The base map was downloaded from the U.S. Geological Survey (https://apps.nationalmap.gov/viewer/).

regions where salinity is below 6 (Fig 2A) [24] and (2) wetlands are associated with improved foraging success of Delta Smelt [15, 67, 68], Delta Smelt in 2011 year-class might have been getting benefits from increased access to tidal wetlands, both in the Suisun Marsh/Montezuma Slough and at the confluence of the rivers in the fall 2011, while fish in the 2013 and 2014 year-classes had access to only the wetlands at the confluence of the Sacramento and San Joaquin rivers and smaller wetlands further upstream [7, 11, 12]. Wetlands used to be abundant in the Delta. In the 1800s, the Delta was a vast 3,000 km$^2$ complex mainly consisting of tule marsh, forested islands, and meandering channels [69, 70]. Over 95% of the wetlands have been lost due to conversion to agriculture and urban development and currently very little wetlands remain [70, 71]. There are activities for restoring wetlands and tidal marshes in the SFE-Delta to improve habitat conditions for Delta Smelt by enhancing food production and availability [72], however the process is challenging since it requires reconciliation with anthropogenic activities.

Elevated water temperature during the dry and critically dry years may have lowered available energy for reproduction. Ectotherms dedicate more energy to respiration at warmer temperatures so that an increase in temperature results in higher metabolic rates, and subsequently an increase in the energy requirements [73]. This can exacerbate nutritional

stress in food limited aquatic ecosystems such as the SFE-Delta [28]. The mean water temperature in the 2014 drought year was approximately 2.6°C higher than 2011 throughout the year with highest water temperature 23.1°C recorded in July 2014 (S4 Fig). Delta Smelt exhibited sub-lethal physiological effects (increase in metabolic rates measured by changes in oxygen consumption) at 20°C [33]. In addition, Komoroske et al. [74] reported that the critical thermal maximum, which is defined as the upper temperature at which fish lose the ability to escape conditions that will ultimately lead to death, of adult Delta Smelt was 27-28°C with continuous feeding. However, it is unknown how Delta Smelt would respond to sub-lethal but potentially stressful temperatures under food-limited conditions. Currently there are only a few publications reporting impacts of elevated water temperature and food limitation on fish reproduction. Donelson et al. [75] reported that coral reef Damselfish (*Acanthochromis polyacanthus*) failed to reproduce when cultured at higher water temperatures with a lower quantity of food. It is very likely that responses to the multiple stressors (i.e., warmer water temperature and food limitation) can vary among different fish species, therefore further analyses are warranted to better understand the impacts of elevated water temperature on metabolic demand and subsequently on growth and reproduction of Delta Smelt under food limited conditions. Whatever the cause of the poor reproductive indices during the drought, Delta Smelt abundances reached historical lows as the drought peaked in 2015–2016 [68, 76].

It is noteworthy that no difference was detected in somatic condition factor among the 2011–2014 year-classes while significant differences were observed in the reproductive endpoints (Fig 5). This can be explained by energy allocation for somatic growth over reproduction, more specifically, Delta Smelt may use energy preferentially for their somatic growth and survival, and then use 'surplus energy' for gonadal development. Strategies of energy allocation for reproduction can vary depending on fish species, however species can be largely classified as 'capital breeders' or 'income breeders'. Capital breeders show cessation or reduction in feeding activity during spawning season and use stored energy for reproduction, while income breeders continue feeding throughout the spawning season and use energy from food intake for their survival and reproduction [77]. Sockeye Salmon (*Oncorhynchus nerka*) is a typical capital breeder because it ceases feeding during spawning season and uses reserved energy for reproduction. In contrast, income breeders prioritize their survival over spawning [78]. Income breeders are often observed in multiple spawners such as Medaka (*Oryzias latipes*) and Rare Minnow (*Gobiocypris rarus*). When mature Rare Minnow are starved, late maturing oocytes start to degrade and are reabsorbed [78]. Therefore food availability for adults is critically important for income breeders for their reproduction. Although Delta Smelt is a multiple spawner as indicated by the presence of immature oocytes in the gonad of fully mature females by histology [38], Delta Smelt could be a capital breeder since (1) reduction in feeding activity was observed prior to and during the spawning season in the wild and at a fish culture facility [28, 79] and (2) atrophic oocytes are rarely found in wild and cultured fish by histology (Personal observation). Further investigation is needed since the reduction in feeding activity during the spawning season in the wild may be because of other causes such as food availability or abiotic factors such as lower water temperatures. However, if Delta Smelt is a capital breeder, accessibility to abundant prey items during pre-spawning season at the subadult stage (fall and early winter) would be critically important to increase 'surplus energy' which can be used for gonadal development.

Warmer water temperature during the drought appeared to have affected timing of maturation. Delta Smelt in the 2013 and 2014 year-classes matured earlier compared with the 2011 year-class, with a prominent difference in February (Fig 4). A possible explanation of the earlier maturation during the drought is the warmer water temperature in January, when the fish in the 2013 and 2014 year-classes experienced approximately 1.0°C higher water temperature than the 2011 year-class (S4 Fig). As Damon et al. [32] reported, elevated water temperature

shifted the spawning season earlier. The early maturation may provide a longer spawning window which could result in higher spawning frequency assuming that energy is not limiting, however it is unlikely under the current food-limited conditions during the summer and fall [28, 80]. Increased temperatures later in the spawning season can also cease spawning early, resulting in a shortened spawning window [32, 81]. This can be especially problematic in years where clutch fecundity size is lower, such as 2013 and 2014. Reduced annual fecundity in Delta Smelt is therefore a concern given climate change predictions of warmer and drier conditions in the California [30, 31].

## Conclusions

In this study we report on (1) the distribution pattern of female Delta Smelt during their winter and spring spawning season and (2) the impacts of severe drought on their reproductive performance. Salinity better explained the distribution pattern of Delta Smelt at subadult and adult stages compared with water temperature or turbidity. Although there are some freshwater locations where mature Delta Smelt can be frequently found during the spawning season (e.g., Cache Slough Complex and Suisun Marsh/Montezuma Slough), Delta Smelt at the final maturation stage (Stage 5: hydration) appeared to be widespread mainly in the areas where salinity was below 1.0 during the spawning season. Therefore, Delta Smelt could theoretically spawn in any freshwater locations, with more specific spawning requirements in the wild (e.g., substrate type and depth) still unknown. Delta Smelt, which experienced dry and drought conditions (2013 and 2014 year-classes), had smaller oocytes and lower clutch sizes and gonadosomatic index than the fish caught in a wet year (2011 year-class), suggesting reproductive performance was negatively affected by environmental conditions during the drought.

## Supporting information

**S1 Fig. Turbidity at each reproductive stage of female Delta Smelt from the 2011–2014 year-classes including extreme values (>150 NTU).** Maturity of female fish was scored based on the gonadal histological features [38].
(TIFF)

**S2 Fig. Maturity and distribution pattern of female Delta Smelt for 2001–2010 year-classes.** The data were obtained from the California Department of Fish and Wildlife website (https://www.wildlife.ca.gov/Conservation/Delta/Spring-Kodiak-Trawl). The blue and red lines indicate salinity boundary for 1.0 and 6.0, respectively.
(TIFF)

**S3 Fig. Maturity and distribution pattern of female Delta Smelt for the 2014 year-class.**
(TIFF)

**S4 Fig. Comparison of water temperatures that Delta Smelt (2011–2014 year-classes) likely experienced in the wild, from spring for larvae (April-June), summer for juveniles (June-September), fall for subadults (September-December), and winter for adults (January-April).** The water temperature data were obtained from the California Department of Fish and Wildlife FTP server (ftp://ftp.wildlife.ca.gov/; 20mm Survey, Fall Midwater Trawl Survey, and Spring Kodiak Trawl Survey). The August data are not available. Mean water temperatures of the four regions (Suisun Marsh and Montezuma Slough, Confluence of the Sacramento and San Joaquin rivers, Sacramento River, and Cache Slough Complex) are depicted for each month. Stations with salinity higher than 6 are not included. The error bars indicate standard deviation.
(TIFF)

**S1 Table. A summary of Spring Kodiak Trawl for 2011 through 2014 year-classes.** Numbers in the table indicate number of stations that field sampling was performed.
(XLSX)

**S2 Table. The dataset used for the analyses in this study, including water quality parameters (temperature, salinity, and turbidity) and fish somatic and reproductive data (fork length, total weight, condition factor, gonadosomatic index, maturity, area of oocytes, and more).**
(CSV)

## Acknowledgments

We are grateful to Ching Teh, Alireza Javidmehr, Chris Perry, Tena Dhayalan, Franklin Tran, Gary Wu, Georgia Ramos, and other members at the Aquatic Health Program, UC Davis, for dissecting and processing fish for obtaining the data used in this study. The authors also thank Randall D. Baxter and other staff members at the California Department of Fish and Wildlife for field sampling and for maintaining historical data and Nicholas Bertrand at the United States Bureau of Reclamation for reviewing the manuscript. The views expressed are those of the authors and do not represent the official opinion of any employer, institution or government agency.

## Author Contributions

**Conceptualization:** Tomofumi Kurobe, Bruce G. Hammock, Tien-Chieh Hung, Andrew A. Schultz, Swee J. Teh.

**Data curation:** Tomofumi Kurobe, Bruce G. Hammock, Lauren J. Damon, Shawn Acuña.

**Formal analysis:** Tomofumi Kurobe, Bruce G. Hammock, Swee J. Teh.

**Funding acquisition:** Tomofumi Kurobe, Andrew A. Schultz, Swee J. Teh.

**Investigation:** Tomofumi Kurobe.

**Methodology:** Tomofumi Kurobe, Bruce G. Hammock, Lauren J. Damon, Tien-Chieh Hung, Shawn Acuña, Swee J. Teh.

**Project administration:** Andrew A. Schultz, Swee J. Teh.

**Resources:** Lauren J. Damon, Tien-Chieh Hung, Swee J. Teh.

**Supervision:** Swee J. Teh.

**Validation:** Tomofumi Kurobe.

**Visualization:** Tomofumi Kurobe.

**Writing – original draft:** Tomofumi Kurobe.

**Writing – review & editing:** Tomofumi Kurobe, Bruce G. Hammock, Lauren J. Damon, Tien-Chieh Hung, Shawn Acuña, Andrew A. Schultz, Swee J. Teh.

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
