## [Decision Letter · Decision Letter 0]

7 Dec 2021

PONE-D-21-30888Reproductive strategy of Delta Smelt Hypomesus transpacificus and impacts of drought on reproductive performancePLOS ONE

Dear Dr. Kurobe,

Thank you for submitting your manuscript to PLOS ONE. After careful consideration, we feel that it has merit but does not fully meet PLOS ONE’s publication criteria as it currently stands. Therefore, we invite you to submit a revised version of the manuscript that addresses the points raised during the review process.

We look forward to receiving your revised manuscript.

Kind regards,

Ram Kumar, Ph.D.

Academic Editor

PLOS ONE

Journal Requirements:

2. To comply with PLOS ONE submissions requirements, please provide methods of sacrifice (which substances and/or methods were applied (please also indicate if anesthesia was used at any point and if so what substances and/or methods were applied)), and if the method of sacrifice is part of routine survey in the Methods section of your manuscript.

3. In your Methods section, please provide additional information regarding the permits you obtained for the work. Please ensure you have included the full name of the authority that approved the field site access and endangered species sampling. If no permits were required please include a brief statement explaining why.

4. In your Methods section, please provide additional location information, including geographic coordinates of your field collection site if available.

“This work was supported by a grant to SJT and TK (U.S.Bureau of Reclamation R17AC00129). Partial support was provided by grants to SJT from the California Department of Fish and Wildlife Ecosystem Restoration Program E1183004 and U.S. Geological Survey G12AC20079 and G15AS00018 (Erwin Van Nieuwenhuyse program manager).”

8. We note that Figure 1, 3 and 6 in your submission contain map images which may be copyrighted. All PLOS content is published under the Creative Commons Attribution License (CC BY 4.0), which means that the manuscript, images, and Supporting Information files will be freely available online, and any third party is permitted to access, download, copy, distribute, and use these materials in any way, even commercially, with proper attribution. For these reasons, we cannot publish previously copyrighted maps or satellite images created using proprietary data, such as Google software (Google Maps, Street View, and Earth). For more information, see our copyright guidelines: http://journals.plos.org/plosone/s/licenses-and-copyright.

 a. You may seek permission from the original copyright holder of Figure 1, 3 and 6 to publish the content specifically under the CC BY 4.0 license. 

Additional Editor Comments:

Dear Dr. Tomofumi Kurobe

Thank you for considering the PLOsOne journal for your paper. We have got your manuscript reviewed by four different reviewers. All the reviewers appreciate the manuscript and recommend publication. The manuscript will be published after minor revision . Therefore I would request you to comply with the reviewers comments and resubmit the manuscript after suggested revision. Following comments could not be uploaded by the reviewers rather they sent to me as email attachment.

Reviewer : Rai, Malayaj

Title : “Reproductive strategy of Delta Smelt Hypomesus transpacificus and impacts of drought on reproductive performance”

1. Recommendation (Accept, Minor Revision, Major Revision, Reject)

Accept

2. Is the manuscript technically sound, and do the data support the conclusions? (Answer options: Yes, No, Partly)

Yes

3. Has the statistical analysis been performed appropriately and rigorously? (Answer options: Yes, No, I don't know, N/A)

Yes

4. Does the manuscript adhere to the PLOS Data Policy? Additional details can be found at http://www.plosone.org/static/policies#sharing

Yes

5. Is the manuscript presented in an intelligible fashion and written in standard English?

Yes

6. Review Comments to the Author

This study aims to describe the distribution of delta-smelt during spawning and influence of drought on its reproductive efficiency. The first hypothesis is not confirmed as the distribution of delta smelt seems to be widespread in regions with low salinity (below 1.0), which is indicated in the figures provided by the author and also discussed in the paper. The second hypothesis regarding the impact of drought on reproductive efficacy of delta-smelt could be confirmed as indicated by a decrease in clutch size in 2013 and 2014 year-classes in comparison to 2011 year-class. Decrease in GSI post-drought is also indicating a negative impact of drought on reproductive performance. The statistical analysis method is correctly described and the reason for the choices tests is also articulately provided such as why the tests were performed on fish in the late vitellogenic stage which makes it easier to understand for reviewers from different areas of interest or expertise. Overall, the writing is intelligible and informative. The reason for current distribution or a more detailed analysis as what were the causative agents for a reduced reproductive need to be studied in more detail.

7. Would you like your identity revealed to the authors of this submission? (Answer options: Yes, No)

Yes

8. Do you have any potentially competing interests? If none, type "None." Our policy on competing interests can be found at http://www.plosone.org/static/policies.action#competing.

No

 

Manuscript #: PONE-D-21-30888

Title: Reproductive strategy of Delta Smelt Hypomesus transpacificus and impacts of drought on reproductive performance

Article type: Research Article

Authors: Tomofumi Kurobe; Bruce Hammock; Lauren Damon; Tien-Chieh Hung; Shawn Acuña; Erwin Van Nieuwenhuyse; Andrew Schultz; Swee The

1. Recommendation (Accept, Minor Revision, Major Revision, Reject)

Minor revision

2. Is the manuscript technically sound, and do the data support the conclusions? (Answer options: Yes, No, Partly)

Yes

3. Has the statistical analysis been performed appropriately and rigorously? (Answer options: Yes, No, I don't know, N/A)

Yes

4. Does the manuscript adhere to the PLOS Data Policy? Additional details can be found at http://www.plosone.org/static/policies#sharing

Yes

5. Is the manuscript presented in an intelligible fashion and written in standard English? (Answer options: Yes, No)

Yes

6. Review Comments to the Author

The present study intends to fill the research gaps involved in Delta smelt distribution at the time of spawning and the impact of drought on it. The language of the paper is expressible and is written in an explainable manner, however, the use of frequent passive voice sentences should be done away with. Instead, as suggested, the authors should adhere to active voice, unless needed otherwise.

In lines 130-131, the sentence runs as “Another unknown is whether a severe, recent drought in California from 2013- 131 2015 impacted the reproductive performance of Delta Smelt. Feyrer et al. The word “unknown” should be followed by a noun such as “fact”. It would become more impressive if the author uploads the data involved in the research.

7. Would you like your identity revealed to the authors of this submission? (Answer options: Yes, No)

No

8. Do you have any potentially competing interests? If none, type "None."

None

Reviewers' comments:

Reviewer's Responses to Questions

**Comments to the Author**

1. Is the manuscript technically sound, and do the data support the conclusions?

Reviewer #1: Yes

Reviewer #2: Yes

2. Has the statistical analysis been performed appropriately and rigorously? 

Reviewer #1: Yes

Reviewer #2: Yes

3. Have the authors made all data underlying the findings in their manuscript fully available?

Reviewer #1: No

Reviewer #2: Yes

4. Is the manuscript presented in an intelligible fashion and written in standard English?

Reviewer #1: Yes

Reviewer #2: Yes

5. Review Comments to the Author

Reviewer #1: The study aims to address the research gap on Delta Smelt distribution during spawning and the impact of drought on its reproductive performance. The methods employed appropriate design, including the choice of statistical tests. The results answered the hypotheses of the study. However, the discussion at lines 595-598 needs to be cautioned from overinterpreting the water quality data used in the study. From an ecological perspective, chlorophyll may serve as a more appropriate indicator for the spatio-temporal distribution in the context of small pelagic fisheries of the Zamboanga Peninsula (Villanoy et al., 2014). Perhaps the paper can expand its discussion to include the role of chlorophyll insofar as Delta Smelt distribution is concerned.

On the other hand, the impact of drought adequately correlates with the reduction of oocyte size, clutch size, and GSI; the data strongly supports the study's second hypothesis. Regarding PLOS data policy compliance, can the authors upload the data points for somatic condition factor, clutch size, oocyte area, and other related data in the study? For this reason, I answered "no" to review question 3. The writing style of the paper is intelligible and written in standard English.

Reviewer #2: Good study, provide new knowledge of female Delta Smelt relating to environmental factors.

6. PLOS authors have the option to publish the peer review history of their article (what does this mean?). If published, this will include your full peer review and any attached files.

Reviewer #1: **Yes: **Robert S. Guino-o II

Reviewer #2: **Yes: **Kareem Altaff

---

## [Author Response · Author response to Decision Letter 0]

6 Feb 2022

Additional Editor Comments:

Dear Dr. Tomofumi Kurobe

Thank you for considering the PLOsOne journal for your paper. We have got your manuscript reviewed by four different reviewers. All the reviewers appreciate the manuscript and recommend publication. The manuscript will be published after minor revision . Therefore I would request you to comply with the reviewers comments and resubmit the manuscript after suggested revision. Following comments could not be uploaded by the reviewers rather they sent to me as email attachment.

Reviewer : Rai, Malayaj

Title : “Reproductive strategy of Delta Smelt Hypomesus transpacificus and impacts of drought on reproductive performance”

1. Recommendation (Accept, Minor Revision, Major Revision, Reject)

Accept

2. Is the manuscript technically sound, and do the data support the conclusions? (Answer options: Yes, No, Partly)

Yes

3. Has the statistical analysis been performed appropriately and rigorously? (Answer options: Yes, No, I don't know, N/A)

Yes

4. Does the manuscript adhere to the PLOS Data Policy? Additional details can be found at http://www.plosone.org/static/policies#sharing

Yes

5. Is the manuscript presented in an intelligible fashion and written in standard English?

Yes

6. Review Comments to the Author

This study aims to describe the distribution of delta-smelt during spawning and influence of drought on its reproductive efficiency. The first hypothesis is not confirmed as the distribution of delta smelt seems to be widespread in regions with low salinity (below 1.0), which is indicated in the figures provided by the author and also discussed in the paper. The second hypothesis regarding the impact of drought on reproductive efficacy of delta-smelt could be confirmed as indicated by a decrease in clutch size in 2013 and 2014 year-classes in comparison to 2011 year-class. Decrease in GSI post-drought is also indicating a negative impact of drought on reproductive performance. The statistical analysis method is correctly described and the reason for the choices tests is also articulately provided such as why the tests were performed on fish in the late vitellogenic stage which makes it easier to understand for reviewers from different areas of interest or expertise. Overall, the writing is intelligible and informative. The reason for current distribution or a more detailed analysis as what were the causative agents for a reduced reproductive need to be studied in more detail.

Thank you for the comment, especially about further study to fill the gap between reduced reproductive performance and abiotic factors described in the paper. We are planning to do it in the future study.

7. Would you like your identity revealed to the authors of this submission? (Answer options: Yes, No)

Yes

8. Do you have any potentially competing interests? If none, type "None." Our policy on competing interests can be found at http://www.plosone.org/static/policies.action#competing.

No

 

Manuscript #: PONE-D-21-30888

Title: Reproductive strategy of Delta Smelt Hypomesus transpacificus and impacts of drought on reproductive performance

Article type: Research Article

Authors: Tomofumi Kurobe; Bruce Hammock; Lauren Damon; Tien-Chieh Hung; Shawn Acuña; Erwin Van Nieuwenhuyse; Andrew Schultz; Swee The

1. Recommendation (Accept, Minor Revision, Major Revision, Reject)

Minor revision

2. Is the manuscript technically sound, and do the data support the conclusions? (Answer options: Yes, No, Partly)

Yes

3. Has the statistical analysis been performed appropriately and rigorously? (Answer options: Yes, No, I don't know, N/A)

Yes

4. Does the manuscript adhere to the PLOS Data Policy? Additional details can be found at http://www.plosone.org/static/policies#sharing

Yes

5. Is the manuscript presented in an intelligible fashion and written in standard English? (Answer options: Yes, No)

Yes

6. Review Comments to the Author

The present study intends to fill the research gaps involved in Delta smelt distribution at the time of spawning and the impact of drought on it. The language of the paper is expressible and is written in an explainable manner, however, the use of frequent passive voice sentences should be done away with. Instead, as suggested, the authors should adhere to active voice, unless needed otherwise.

Thank you for the suggestion. However, we would like to stick with as it is. Active voice was carefully used to emphasize messages that we want to deliver to readers.

In lines 130-131, the sentence runs as “Another unknown is whether a severe, recent drought in California from 2013- 131 2015 impacted the reproductive performance of Delta Smelt. Feyrer et al. The word “unknown” should be followed by a noun such as “fact”. It would become more impressive if the author uploads the data involved in the research.

I cannot fully agree with the idea of adding “fact” or other relevant words since we didn’t have any “fact” or “evidence” when we generated the hypothesis. It was simply a “unknown”.

We uploaded the data generated from the study.

7. Would you like your identity revealed to the authors of this submission? (Answer options: Yes, No)

No

8. Do you have any potentially competing interests? If none, type "None."

None

Reviewers' comments:

Reviewer's Responses to Questions

Comments to the Author

1. Is the manuscript technically sound, and do the data support the conclusions?

Reviewer #1: Yes

Reviewer #2: Yes

2. Has the statistical analysis been performed appropriately and rigorously?

Reviewer #1: Yes

Reviewer #2: Yes

3. Have the authors made all data underlying the findings in their manuscript fully available?

Reviewer #1: No

Reviewer #2: Yes

4. Is the manuscript presented in an intelligible fashion and written in standard English?

Reviewer #1: Yes

Reviewer #2: Yes

5. Review Comments to the Author

Reviewer #1: The study aims to address the research gap on Delta Smelt distribution during spawning and the impact of drought on its reproductive performance. The methods employed appropriate design, including the choice of statistical tests. The results answered the hypotheses of the study. However, the discussion at lines 595-598 needs to be cautioned from overinterpreting the water quality data used in the study. From an ecological perspective, chlorophyll may serve as a more appropriate indicator for the spatio-temporal distribution in the context of small pelagic fisheries of the Zamboanga Peninsula (Villanoy et al., 2014). Perhaps the paper can expand its discussion to include the role of chlorophyll insofar as Delta Smelt distribution is concerned.

We are interested in the role of chl-a on fish health and how it affects distribution patterns of fishes, but I hesitate to include the component in the discussion section. It’s because we didn’t generate our hypotheses to answer the question. I understand that it’s a very interesting topic and, actually, we started some experiment to address the question and getting some interesting data. We are hoping to publish a paper about the topic in near future.

On the other hand, the impact of drought adequately correlates with the reduction of oocyte size, clutch size, and GSI; the data strongly supports the study's second hypothesis. Regarding PLOS data policy compliance, can the authors upload the data points for somatic condition factor, clutch size, oocyte area, and other related data in the study? For this reason, I answered "no" to review question 3. The writing style of the paper is intelligible and written in standard English.

We uploaded data as a csv file.

Reviewer #2: Good study, provide new knowledge of female Delta Smelt relating to environmental factors.

6. PLOS authors have the option to publish the peer review history of their article (what does this mean?). If published, this will include your full peer review and any attached files.

Do you want your identity to be public for this peer review? For information about this choice, including consent withdrawal, please see our Privacy Policy.

Reviewer #1: Yes: Robert S. Guino-o II

Reviewer #2: Yes: Kareem Altaff

---

## [Editor Report · Decision Letter 1]

16 Feb 2022

Reproductive strategy of Delta Smelt Hypomesus transpacificus and impacts of drought on reproductive performance

PONE-D-21-30888R1

Dear Dr. Kurobe

We’re pleased to inform you that your manuscript has been judged scientifically suitable for publication and will be formally accepted for publication once it meets all outstanding technical requirements.

Kind regards,

Ram Kumar, Ph.D., D. Sc (H/C)

Academic Editor

PLOS ONE

Additional Editor Comments (optional):

Thank you for considering the PLoS ONE for your research outcome entitled “Reproductive strategy of Delta Smelt Hypomesus transpacificus and impacts of drought on reproductive performance” . Based on comments of all five reviewers and my own perusal of the Ms. I am glad to inform you that the manuscript is accepted for the publication in the Journal PLoS ONE.

We hope that you will find the PLoS ONE a potential vehicle of disseminating you research in future also.

---

## [Editor Report · Acceptance letter]

1 Mar 2022

PONE-D-21-30888R1 

Reproductive strategy of Delta Smelt *Hypomesus transpacificus* and impacts of drought on reproductive performance 

Dear Dr. Kurobe:

I'm pleased to inform you that your manuscript has been deemed suitable for publication in PLOS ONE. Congratulations! Your manuscript is now with our production department. 

Kind regards, 

on behalf of

Professor Ram Kumar 

Academic Editor

PLOS ONE